# Open Technical Problems in Open-Weight AI Model Risk Management

**Stephen Casper** *MIT CSAIL*                                                                 *scasper@mit.edu*

**Kyle O'Brien** *ERA Fellowship*

**Shayne Longpre** *MIT*

**Elizabeth Seger** *Demos*

**Kevin Klyman** *Stanford University*

**Rishi Bommasani** *Stanford University*

**Aniruddha Nrusimha** *MIT CSAIL*

**Ilia Shumailov** *AI Sequrity Company*

**Sören Mindermann** *Mila, Université de Montréal LawZero*

**Steven Basart** *Center for AI Safety*

**Frank Rudzicz** *Dalhousie University Vector Institute*

**Kellin Pelrine** *FAR.AI*

**Avijit Ghosh** *Hugging Face*

**Andrew Strait** *UK AI Security Institute*

**Robert Kirk** *UK AI Security Institute*

**Dan Hendrycks** *Center for AI Safety*

**Peter Henderson** *Princeton University*

**Zico Kolter** *Carnegie Mellon University*

**Geoffrey Irving** *UK AI Security Institute*

**Yarin Gal** *UK AI Security Institute OATML, University of Oxford*

**Yoshua Bengio** *Mila, Université de Montréal*

**Dylan Hadfield-Menell** *MIT CSAIL*

**Reviewed on OpenReview:** *https://openreview.net/forum?id=8QyGLnFkzc*

## Abstract

Frontier AI models with openly available weights are steadily becoming more powerful and widely adopted. However, compared to proprietary models, open-weight models pose different opportunities and challenges for effective risk management. For example, they allow for more open research and testing. However, managing their risks is also challenging because they can be modified arbitrarily, used without oversight, and spread irreversibly. Currently, there is limited research on safety tooling specific to open-weight models. Addressing these gaps will be key to both realizing their benefits and mitigating their harms. In this paper, we present 16 open technical challenges for open-weight model safety involving training data, training algorithms, evaluations, deployment, and ecosystem monitoring. We conclude by discussing the nascent state of the field, emphasizing that openness about research, methods, and evaluations – not just weights – will be key to building a rigorous science of open-weight model risk management.

# 1 Introduction

Open-weight AI models – models whose weights are publicly available to download – have quickly grown in their capabilities and prominence (Cottier, 2024; Bhandari et al., 2025). 2025 has been a major year for advanced open language, image, and video models (see Table 2 for examples). Simultaneously, proprietary model developers have reported that their models are approaching critical risk thresholds (Google AI, 2025; Anthropic, 2025; OpenAI, 2025). Meanwhile current work estimates that the capabilities of frontier open-weight models only lag 6 to 12 months behind (Cottier, 2024; Maslej et al., 2025). This suggests that open-weight models could soon cross critical capability thresholds.

Open- versus closed-weight deployments come with different safety tradeoffs. While open-weight models allow for more open research and testing, they also come with a greater potential for misuse. Frontier closed weight model developers often rely on a complex combination of training interventions, content classifiers, and enforcement of acceptable use policies to reduce risks (e.g., Google AI, 2025; Anthropic, 2025; OpenAI, 2025). However, none of these methods provide reliable assurances for open-weight models, which can be used, tampered with, and distributed without limitations. Compared to closed-weight models, the attack surface for open-weight models is larger, and the toolkit of reliable techniques for defending them is less well studied. Furthermore, as increasing numbers of open models are released and shared (Bhandari et al., 2025), it is difficult to understand the extent of their spread, usage, and impacts.

Building the field of technical safeguards for open models will be key to capturing their benefits and minimizing their risks. In Section 2, we expand on difficulties, highlighting how *tampering threats* and the *diffuse spread of open models* are defining challenges for managing their risks. Section 3 then outlines technical objectives targeting these gaps. We organize them into five categories spanning the model lifecycle. Objectives 1-3 address threats to open-weight models from harmful tampering, while 3-5 focus on improving access to actionable information about the real-world uses and risks of open models:

1. *Training data curation* methods for preventing models from learning harmful capabilities (Section 4.1). Recent research has shown that these methods are effective at making open-weight models resist few-shot learning harmful behaviors.

2. *Tamper-resistant training and 'unlearning' algorithms* for building additional defenses against malicious fine-tuning and other forms of tampering (Section 4.2).

3. *Model tampering evaluations* for testing model risks under misuse threats from harmful tampering (Section 4.3). These methods are necessary to evaluate real-world risks from downstream model modifications to open-weight models.

4. *Staged deployment strategies* which allow developers to experiment with partial access before a full open release (Section 4.4). These techniques allow developers to monitor for unexpected uses and modify their plans for safeguards and release before a model is made openly available.

5. *Model provenance and forensics* strategies for monitoring real-world uses and impacts (Section 4.5). These strategies offer tools for developers, academics, and other stakeholders to study the diffuse open-weight model ecosystem.

# 2 Why is open-weight model risk management challenging?

Open-weight models can be used and adapted widely without centralized control. This is key to their benefits, enabling more widespread research and decentralization of power (Bommasani et al., 2024; Telecommunications & Administration, 2024; Seger, 2024; Eiras et al., 2024b; Kapoor et al., 2024; François et al., 2025; Longpre et al., 2025b; Miller et al., 2025). For example, the release of DeepSeek R1 has enabled independent safety research of near-frontier models (Goodfire, 2025; Zhou et al., 2025a). However, these same characteristics also make open-weight model risk management distinctly challenging.

**Users can disable safety tools that are external to the model.** A key strategy for managing frontier AI risks is to augment models with external safeguards to monitor for signs of risk and intervene to prevent

harm (Bengio et al., 2025b;a; Sharma et al., 2025; Korbak et al., 2025). For example, it is common for AI models to be deployed with input and/or output filters designed to detect and block harmful uses. These kinds of tools can be very valuable to release alongside open-weight models, but they are also trivial for users with access to the model to disable.

**Downstream users can *tamper* with open models via fine-tuning or other modifications to remove safeguards or add harmful capabilities.** Open and closed-weight AI models alike can both be vulnerable to jailbreaks or other adversarial prompts that elicit harmful behavior. However, open-weight models are additionally prone to powerful tampering threats. Benign fine-tuning, adversarial fine-tuning, and other modifications to a model have been shown to effectively elicit harmful behaviors and capabilities – even from models that are relatively safe off the shelf (e.g., Qi et al., 2023; Hu et al., 2025b; Greenblatt et al., 2024; Wei et al., 2024; Hofstätter et al., 2025; Che et al., 2025).[1] There is growing precedent for open-weight models being fine-tuned and shared specifically for harmful uses. For example, modified open-weight diffusion models have become the most common tools used for creating synthetic child sexual abuse material (IWF, 2024; Hawkins et al., 2025; Vaughan). Recent research has also identified open-weight model variants that have been fine-tuned specifically to perform malicious tasks (Simonovich, 2025). Meanwhile, thousands of open-weight text models have been specifically fine-tuned to disable safeguards. With enough fine-tuning on enough data, safeguards for any model can be undone, meaning that practical anti-tampering techniques can only hope to make harmful forms of fine-tuning sufficiently onerous.

**Open-weight models can be spread quickly and irreversibly.** If a closed-weight model is found to pose hazards, risk-conscious developers can add patches or pull the model from distribution. Consider, for example, OpenAI's April 2025 update of GPT-4o. After release, external evaluation identified excessive sycophancy and encouragement of self-harm. In response, OpenAI reverted to a previous version of the model (OpenAI, 2025b). In contrast, OpenAI's open-weight release of gpt-oss-120b, which currently has over 3 million monthly downloads from HuggingFace, was not reversible. While ceasing service to a model can make it much less accessible (e.g., Solaiman et al., 2025; Seger, 2024), there is no reliable way to prevent existing copies of the model from being used and shared.

**Open-weight models cannot be centrally monitored or moderated.** When closed-weight models are released, they are generally made available through an API controlled by the model deployer. This allows for the developer to use 'Know Your Customer' strategies (Jami Pour et al., 2024), monitor for misuse (OpenAI, 2025a; Yueh-Han et al., 2025; Brown et al., 2025), and enforce acceptable usage policies. In contrast, open-weight models generally cannot be centrally monitored.

**Open-weight models have more complex supply chains than closed models.** Model supply chains involve many resources, including talent, data, compute, and infrastructure (Cen et al., 2023; Longpre et al., 2023a). Closed-weight models can be developed by multiple actors. However, they tend to be developed in a relatively centralized and coordinated way. In contrast, open-weight models more often result from many stages of modification and redistribution, often across jurisdictions. Models with complex supply chains are more prone to single actors introducing harmful behaviors (including backdoors, Hanif et al., 2025), and they make it more difficult to determine accountability for harms (Nissenbaum, 1996; Cooper et al., 2022).

**It can be difficult to track the real-world spread, usage, and impacts of open-weight models.** Because their usage is typically much less centralized, it is often hard for researchers to thoroughly understand the spread, uses, and impact of open-weight models. This makes it more difficult to study risks, perform cost-benefit analysis, and identify effective points of intervention in the open-weight model ecosystem.

# 3 The Toolkit

## 3.1 Technical Safeguards for Open-Weight models

**Scope and relation to prior work:** This paper builds on prior research on open-weight models and their implications for risk management. This includes prior work on outlining the risks and benefits of highly capable open-weight models (Chan et al., 2023; Seger et al., 2023; Eiras et al., 2024b; Kapoor et al., 2024;

---

[1] Notably, several of these papers cited here demonstrated model vulnerabilities to tampering threats via fine-tuning APIs.

Bengio et al., 2025b;b), proposals for risk management frameworks (Liang et al., 2022; Longpre et al., 2025a; Gal & Casper, 2025), and considerations for governance (Bommasani et al., 2024; Telecommunications & Administration, 2024). In particular, our work is closely related to Seger (2024), François et al. (2025), Srikumar et al. (2024), and UK AISI (2025), which each taxonomize and overview approaches for managing risks from open-weight AI models. To complement these past works, **we focus only on open problems for technical safeguards[2] that have distinct implications[3] for open-weight models.**

**Taxonomizing technical safeguards with distinct applicability to open-weight model risk management:** Based in part on taxonomies provided in Seger (2024), François et al. (2025), Srikumar et al. (2024), and UK AISI (2025), we organize technical safeguards with distinct implications for open-weight model risk management in Table 1. Our taxonomy includes five categories corresponding to different stages of a model's lifecycle: training data curation, training, evaluation, deployment, and post-deployment monitoring.

| Approach | Seger (2024) | François et al. (2025) | Srikumar et al. (2024) | UK AISI (2025) |
|---|:---:|:---:|:---:|:---:|
| **Training Data Curation** (Section 4.1) | ✔ | ✔ | ✔ | ✔ |
| **Tamper-Resistant Training & Unlearning Algs.** (Section 4.2) | ✔ | ✔ | ✔ | ✔ |
| **Model Tampering Evaluations** (Section 4.3) | ✔ | ✘ | ✔ | ✔ |
| **Staged Deployment Strategies** (Section 4.4) | ✔ | ✘ | ✔ | ✔ |
| **Model Provenance & Forensics** (Section 4.5) | ✔ | ✘ | ✔ | ✔ |

Table 1: Our taxonomy of technical safeguards with distinct implications for open-weight models sorts methods into five categories corresponding to the different stages of a model's lifecycle: training data curation, training, evaluation, deployment, and post-deployment monitoring. In this table, we overview these methods' coverage in prior public work.

## 3.2 What this Paper Does Not Cover

The primary goal of this paper is to help build the technical science of open-weight model risk management. As a result, we only focus on open problems for technical safeguards with distinct implications for open-weight models. However, this is not to say that other risk-management strategies are not crucial for managing risks from open-weight models. For the purposes of this paper, the following strategies discussed by Bommasani et al. (2024), Telecommunications & Administration (2024), Seger (2024), François et al. (2025), Srikumar et al. (2024), and UK AISI (2025) are out of scope:

- Technical tools for AI risk management that do not have distinct implications for open-weight models:

    - General (non-tamper-resistant) safety fine-tuning techniques (Seger, 2024; François et al., 2025; Srikumar et al., 2024; UK AISI, 2025)

    - Safety scaffolding and content moderation tools (Seger, 2024; François et al., 2025; Srikumar et al., 2024; UK AISI, 2025)

    - Rigorous black-box evaluations (Seger, 2024; François et al., 2025; Srikumar et al., 2024; Zhu et al., 2025c)

---

[2]We define a "technical safeguard" as a technique which uses methods from machine learning research.

[3]We consider a safeguard to have "distinct implications" for open-weight models if it is applicable for open-weight releases and cannot be trivially disabled. For example, we do not consider filters for harmful outputs to have disinct implications for open-weight model risk management as they are also used for closed-weight models and can be trivially disabled in open-weight models.

- ○ Data provenance/forensics (Bommasani et al., 2024; Seger, 2024; François et al., 2025; Srikumar et al., 2024; UK AISI, 2025; Longpre et al., 2024b)
- ○ Monitoring fine-tuning APIs (see also Halawi et al., 2024; Davies et al., 2025)

- Nontechnical approaches for open-weight model risk management:

  - ○ Having an acceptable use policy and proactively enforcing violations (Seger, 2024; Srikumar et al., 2024)
  - ○ Transparency and documentation of code, methods, and evaluation results (Seger, 2024; Srikumar et al., 2024; UK AISI, 2025)
  - ○ Know-your-customer methods (Seger, 2024; UK AISI, 2025) (see also Jami Pour et al., 2024)
  - ○ Implementing incident reporting policies and infrastructure (Seger, 2024; Srikumar et al., 2024) (see also Cattell et al., 2024; McGregor et al., 2025; Longpre et al., 2025b)
  - ○ Monitoring misuse, unintended uses, and user feedback (Telecommunications & Administration, 2024; Seger, 2024; Srikumar et al., 2024)
  - ○ Pulling and replacing models found to pose hazards (Seger, 2024; UK AISI, 2025)

- Accelerating progress in the development of open-weight models with beneficial societal impacts (Bommasani et al., 2024; Seger, 2024; Telecommunications & Administration, 2024; François et al., 2025)

- Governance strategies that carry distinct implications for the open-weight model ecosystem (Bommasani et al., 2024; Seger, 2024; Telecommunications & Administration, 2024; François et al., 2025)

## 4 Open Technical Problems

Here, following the taxonomy in Table 1 above, we discuss open technical problems in open-weight model risk management spanning the model lifecycle from pretraining to post-deployment ecosystem monitoring. The first three challenges we discuss pertain to open-weight model resistance to harmful fine-tuning and other forms of tampering. Section 4.1 focuses on *training data* curation and its implications for how well a final model resists harmful tampering. Section 4.2 additionally covers *training* algorithms to make models resistant to tampering. Section 4.3 focuses on *evaluating* open-weight models under realistic tampering threats. The final two challenges pertain to how models are released and monitored after they are developed. Section 4.4 discusses strategies for *deploying* models in stages. Finally, Section 4.5 covers tools to facilitate the *post-deployment monitoring* of model uses and impacts in the open-weight ecosystem

### 4.1 Training Data Curation

**Training data curation is increasingly understood as a key intervention for improving model safety both off-the-shelf and under tampering.** Frontier AI models are prone to learning harmful information during training (e.g., Phuong et al., 2024; Anthropic, 2025). An intuitive countermeasure is to prevent models from learning harmful capabilities by minimizing exposure to unsafe training data. This aligns with evidence that models acquire most of their core knowledge during pretraining (Zhou et al., 2023; Raghavendra et al., 2024; Chang et al., 2024), where they are exposed to a diverse corpus of data spanning trillions of tokens (e.g. Yang et al., 2025; Meta AI, 2025; Agarwal et al., 2025). However, once knowledge has been internalized by a model, it is empirically difficult to remove or substantially modify (Jia et al., 2021; Anwar et al., 2024). This suggests that training data interventions, particularly during pretraining, could have the potential to shape the core knowledge, concepts, and propensities of the model towards safer outcomes.

**Existing work:** Curating training data at scale and filtering harmful content (such as text with instructions for performing illegal actions or images/videos depicting child sexual abuse) are widely understood as a key means of training safer AI models (e.g., Nichol et al., 2022; Longpre et al., 2023b; Thorn & All Tech Is Human, 2024; Seger, 2024; Longpre et al., 2025a; Liu et al., 2024c; François et al., 2025; Srikumar et al., 2024). Blocking data from known risky sources, such as websites with a high prevalence of adult and toxic

content, is a common but not standardized practice in dataset design (Soldaini et al., 2024; Penedo et al., 2024; Li et al., 2024a). Several works on open datasets have made significant contributions by releasing their data curation pipelines (Gao et al., 2020; Laurençon et al., 2023; Raffel et al., 2019; Soldaini et al., 2024; Kandpal et al., 2025). Other recent works have begun to study how pretraining data curation can be used to prevent unsafe capabilities (Korbak et al., 2023; Chen et al., 2025c; UK AISI, 2025; O'Brien et al., 2025; Wallace et al., 2025; Lee et al., 2025; Liu et al., 2025b). These works highlight notable successes, limitations, and open questions.

**4.1.1. How does data curation's effectiveness differ across harm categories?** Recent work has shown both successes (Maini et al., 2025; Lee et al., 2025; Chen et al., 2025c; O'Brien et al., 2025; Albalak et al.; Liu et al., 2025b) and limitations (Li et al., 2025b; Wallace et al., 2025; Wei et al., 2025) of data curation in scoping model capabilities. For instance, recent works have demonstrated pretraining filtering's ability to significantly reduce models' knowledge of biorisk-related topics (Chen et al., 2025c; O'Brien et al., 2025). However, Wallace et al. (2025) found an implementation of filtering to be ineffective when applied to gpt-oss. These findings suggest that filtering has the potential to be effective in preventing technical capability, but that it may be sensitive to implementational details. However, precise comparisons are challenging due to model cards often lacking information, such as the amount of data filtered or the amount of compute spent on filtering. Taken together, current works suggest that filtering data related to entire science or engineering domains can build more durable safeguards into models (Lee et al., 2025; O'Brien et al., 2025), while filtering data related to simpler propensities (such as toxicity or refusal of harmful requests) (Maini et al., 2025; Li et al., 2025b) or more niche science topics (Wei et al., 2025) does not. Open questions remain regarding data filtering's ability to limit potentially unsafe capabilities that are closely intertwined with beneficial capabilities, such as offensive hacking and defensive cybersecurity (Barez et al., 2025).

**4.1.2. How can scaling data curation be scaled across languages, modalities, and data/model sizes?** Training data curation at scale is deceptively difficult (Paullada et al., 2021) due to costs (Ngo et al., 2021), filtering errors (Ziegler et al., 2022), degradation of dataset quality (Welbl et al., 2021), the massively multilingual nature of internet text (Kreutzer et al., 2022), biases in content moderation (Welbl et al., 2021; Dodge et al., 2021; Xu et al., 2021; Stranisci & Hardmeier, 2025), and the inherently contextual nature of harmfulness (Lindner & El-Assady, 2022). When curating internet-scale datasets, efficiency, precision, and recall are crucial. Slow feedback loops exacerbate these challenges; ineffective data curation may only become apparent at the end of long training runs, potentially requiring retraining the model from scratch. Regarding efficiency and precision, O'Brien et al. (2025) recently introduced a multi-stage approach to filtering that required less than 1% of the subsequent model's training compute. However, their approach sacrificed efficiency for precision, resulting in many benign documents being filtered. Regarding recall, it is unclear whether larger and more sample-efficient models require increasingly extensive filtering. Developing frameworks that expand the pareto frontier of efficiency, precision, and recall will be key to making training data filtering more competitive (e.g., Chen et al., 2025c).

**4.1.3. What is the relationship between training data contents and emergent model capabilities?** More broadly, the general relationship between model architecture, the content of training data, and emergent capabilities is unclear. Recent work on influence functions (e.g., Grosse et al., 2023), out-of-context reasoning (e.g., Berglund et al., 2023; Treutlein et al., 2024; Hu et al., 2025a), coreset analysis (Pal et al., 2025), and domain-aware scaling (Hamidieh et al., 2025) has suggested a surprising ability of language models to infer generalizable knowledge from constitutive information in training data. This presents important questions. Can unsafe capabilities emerge from benign data approved by data filtering pipelines in practical settings? Relatedly, can filtering data from one domain have substantial unintended effects on model capabilities in another domain? Ultimately, a predictive and practically applicable theory of emergent capabilities in state-of-the-art models remains elusive (Wei et al., 2022; Schaeffer et al., 2025). One approach for continued work can be to study how specific behaviors emerge in simple settings. However, the most directly risk-relevant research will be empirical work to examine how realistic interventions on training data affect frontier models. In particular, understanding if and when models can learn dangerous capabilities from innocuous training data, how much data is required, what quality of data is needed, and how dynamics change with scale will be relevant and actionable for managing risks.

### 4.2 Tamper-Resistant Training and Unlearning Algorithms

**Training algorithms designed to make models resist tampering can further improve safety under harmful modifications.** Aside from training data interventions, there is also a growing body of research focused on post-training defenses for open-weight models. In particular, post-training safeguards designed to resist downstream 'tampering' modifications are a core strategy for mitigating the risks from malicious or negligent downstream use of open-weight models.

**Existing work:** Currently, researchers study safety fine-tuning (e.g., Bai et al., 2022) and "machine unlearning" methods (Gao et al., 2024; Liu et al., 2024b; Barez et al., 2025) as strategies for making models more strongly resist harmful behaviors uses such as assisting a user in illegal activity. However, state-of-the-art fine-tuning and unlearning algorithms have consistently been vulnerable to being undone within dozens of steps of adversarial fine-tuning. While research on defenses often reports model resistance to thousands or tens of thousands of examples of adversarial fine-tuning, to the best of our knowledge, the state of the art for tamper-resistance, as assessed by second-party red-teaming efforts, is only around several hundred steps of adversarial fine-tuning (Qi et al., 2023; Yang et al., 2023; Bhardwaj & Poria, 2023; Li et al., 2024b; Lynch et al., 2024; Huang et al., 2024b; Hu et al., 2025b; Łucki et al., 2025; Peng et al., 2024; Deeb & Roger, 2025; Qi et al., 2024; Che et al., 2025; Dorna et al., 2025). This applies even to methods that have been designed to confer tamper resistance (Łucki et al., 2025; Qi et al., 2024; Che et al., 2025). When techniques do withstand multiple rounds of supervised fine-tuning, it tends to come with major tradeoffs to a model's general knowledge and fluency (Qi et al., 2024; Zhou et al., 2024).

**4.2.1. How do we develop more tamper-resistant unlearning algorithms?** The persistent struggles of tamper-resistant unlearning methods prompt a reassessment of current approaches. Prior methods have involved pruning (Lo et al., 2024; Chapagain et al., 2025), meta-learning (Abdalla et al., 2025; Anonymous, 2025a; Rosati et al., 2025; Perin et al., 2025; Li et al., 2025a; Wang et al., 2025b; Yi et al., 2025), training with specialized objectives (Cao, 2025; Feng et al., 2025), training under tampering (Henderson et al., 2023; Huang et al., 2024c;a; Zheng & Yeh, 2024; Fan et al., 2025a; Cheng et al., 2025a; Zheng et al., 2025; Liu et al., 2025a; Sheshadri et al., 2025; Tamirisa et al., 2025; Sanyal et al., 2025), specially-parameterized updates (Sondej & Yang, 2025), and activation noising (Rosati et al., 2024; Pan et al., 2024; Zou et al., 2024; Tamirisa et al., 2025; Abdalla et al., 2025). Benchmarking work has yet to thoroughly compare all of these types of methods. There may be several opportunities for algorithmic innovation. Some inspiration can be taken from some successes of pretraining data filtering (O'Brien et al., 2025; Liu et al., 2025b). It is possible that running tamper-resistance algorithms for a long time and/or during pretraining could confer stronger tamper resistance. Alternative approaches could attempt to leverage a mechanistic understanding of models that are entirely ignorant about topics (e.g., O'Brien et al., 2025; Liu et al., 2025b) to design more principled training objectives compared to existing ones (e.g. Rosati et al., 2024; Zou et al., 2024; Tamirisa et al., 2025; Abdalla et al., 2025). Finally, while Huang et al. (2024c), Huang et al. (2024a), Sheshadri et al. (2025), Tamirisa et al. (2025), and others have used adversarial methods for tamper resistance, they each only train against a narrow class of tampering attacks. Training models against a more diverse assortment of tampering threats might be able to confer more generalizable tamper resistance. On the other hand, there also might be fundamental limitations for post-training methods' abilities to deeply remove or make inaccessible some types of unwanted knowledge from models. Finally, to ensure the competitiveness of safer models, future research on tamper-resistant fine-tuning will need to prioritize striking a precise balance between the removal of harmful capabilities and degrading benign ones.

**4.2.2. How can we robustly edit model beliefs with minimal side effects?** In contrast to making models ignorant about potentially harmful topics, some researchers have proposed introducing specific incorrect beliefs into language models about hazardous procedures (e.g., Wang et al., 2025a) to prevent the model from generating harmful outputs. For example, teaching a model incorrect information about how to acquire child sexual abuse material could be a complementary approach to both refusal training and unlearning. Belief revision can occur via specific edits to their parameters (e.g., Meng et al., 2023; Geva et al., 2023; Zhang et al., 2024) or fine-tuning (e.g., Wang et al., 2025a; Slocum et al., 2025). However, both approaches currently suffer from challenges of robust generalization (e.g., Wu et al., 2025; Zhong et al., 2024; Slocum et al., 2025), scalability (O'Brien et al., 2025), interference with other interventions (Kolbeinsson et al., 2024), and the ripple effects of belief modification (Cohen et al., 2024; Hase et al., 2024). This sug-

gests useful opportunities to develop benchmarks for surgical knowledge revision, improve scalability, limit side-effects, and demonstrate realistic use cases for mitigating specific risks with knowledge editing.

**4.2.3 Can we develop models that effectively resist retrieving harmful information?** LLMs do not necessarily need to know harmful information to provide it to a user. Models are increasingly being augmented with tools to search, retrieve, and synthesize information from the web (OpenAI, 2025; He et al., 2025). For example, O'Brien et al. (2025) showed that a biothreat-ignorant LLM could still effectively answer biothreat-related questions when given information with the answer in context, such as a textbook or scientific paper. This poses a unique challenge for open-weight model risk management because the standard defenses of refusal, API monitoring, and intervention can be disabled for open-weight models. One open challenge is to practically study the differences between the capabilities of domain-ignorant and domain-competent retrieval-augmented models on complex real-world tasks. Human domain experts are more effective than nonexperts at searching for answers to domain questions on the web, so it is intuitive that the same may apply for language models. Nonetheless, to our knowledge, this has not been directly tested in large models. A second challenge is to develop tamper-resistant safeguards that can defend against tool-augmentation attacks. For example, O'Brien et al. (2025) found that some machine unlearning techniques (Zou et al., 2024) were an effective off-the-shelf defense but were not tamper-resistant. Currently, tamper-resistant safeguards against these attacks remain unaddressed.

## 4.3 Model Tampering Evaluations

**Evaluating models under tampering threats is necessary to assess real-world risks from open models.** Internal and external evaluations of frontier AI models are central to emerging AI governance and risk management frameworks. Because many actors could fine-tune open-weight models with unsafe data (Huang et al., 2024b) or insert backdoors after initial pretraining (Bai et al., 2024; Zhou et al., 2025b), evaluations under these types of threats are necessary to fully assess practical risks.

**Existing work:** Fully assessing the risks of open-weight models requires evaluating them under "tampering" (Gal, 2024; Casper et al., 2024; Che et al., 2025; Wallace et al., 2025; O'Brien et al., 2025) threats from fine-tuning, steering, model editing, pruning, or other interventions. However, many current assessments neglect this possibility. For example, tampering evaluations are not reported on in the technical reports for most frontier open-weight models (see Table 2). Standard procedures to assess these gaps have not yet been established. For example, to our knowledge, gpt-oss has been the only frontier open-weight model where pre-release adversarial fine-tuning evaluations have been reported (Wallace et al., 2025; Agarwal et al., 2025). The current lack of common tampering evaluations creates a risk of both missing harmful uplift potential and incentivizing developers to game evaluations with superficial safeguards.

**4.3.1. How can we develop rigorous benchmarking and evaluation frameworks?** While it is widely understood that the potential risks from open-weight models depend greatly on how easily they can be harmfully tampered with, little tampering evaluation infrastructure exists. Notably, two recent toolkits (Hossain et al., 2025; Dombrowski et al., 2025) have introduced frameworks to evaluate the capabilities of models under a suite of tampering threats. However, they do not fully address challenges with these evaluations, such as sensitivity to different forms of elicitation (Fan et al., 2025b), hyperparameter sensitivity (Qi et al., 2024), the diversity of adversarial attacks (Łucki et al., 2025), or the existence of multiple metrics for measuring tampering attacks (tokens, steps, compute, effort, etc). No framework has yet achieved a degree of threat coverage comparable to the full model tampering toolkit, leading to patchwork evaluations in the field that can be difficult to compare and trust (Huang et al., 2024b; Hossain et al., 2025; Qi et al., 2024; Łucki et al., 2025).

**4.3.2. What modifications should be used for worst-case risk estimation under model tampering?** Current research on evaluating tamper resistance has principally focused on fine-tuning threats (Qi et al., 2023; Yang et al., 2023; Bhardwaj & Poria, 2023; Li et al., 2024b; Lynch et al., 2024; Huang et al., 2024b; Hu et al., 2025b; Łucki et al., 2025; Peng et al., 2024; Deeb & Roger, 2025; Qi et al., 2024; Che et al., 2025). However, other types of interventions have been known to impair the safety of models, including pruning (Wei et al., 2024), low-rank modifications (Wei et al., 2024), latent-space attacks (Bailey et al., 2025), model merging (Hammoud et al., 2024), quantization (Egashira et al., 2024; Chen et al., 2025a), distillation

(Yang et al., 2024; Angell et al., 2025), and backdoor insertion algorithms (Bai et al., 2024; Zhou et al., 2025b). These threats have not yet been studied adversarially alongside tamper-resistant algorithms. More rigorous evaluations of open-weight model risks will require considering the full tampering toolkit for model tampering. In particular, it will be important to study the extent to which simple modifications to models might be able to greatly alter their capabilities and associated risks. For example, few-shot fine-tuning and iterative reasoning have been shown to significantly improve over a model's advertised capabilities under default evaluation conditions (Muennighoff et al., 2025). Despite posing significant risk and uncertainties, these types of "capability overhangs" are not well understood for open-weight models. Finally, it is not even currently clear whether it is possible to build enough tampering resistance into models to impose meaningful barriers to misuse. Even if models resist thousands of steps of fine-tuning (O'Brien et al., 2025; Liu et al., 2025b), performing these tampering attacks may only take minutes and cost tens of dollars. It is unclear the extent to which obtaining training data can serve as a meaningful bottleneck.

**4.3.3. How can we systematically identify effective attacks and defenses?** Exhaustively testing tampering attacks is computationally prohibitive. For instance, even a standard fine-tuning attack can vary in multiple aspects: learning rate, training steps, training algorithm, etc. In particular, a number of recent works have shown how fine-tuning dataset contents has a large impact on the effectiveness of adversarial fine-tuning (Shen et al., 2024; Hsiung et al., 2025; Eiras et al., 2024a; PROJECTION, 2025; Xiao et al., 2025; Hu et al., 2025c; Anonymous, 2025b; Ham et al., 2025; Chen et al., 2025b; He et al., 2024). Safety measures that appear robust to some tampering threats can fail against others (Qi et al., 2024; Łucki et al., 2025; Che et al., 2025). Currently, there is not yet a general understanding of which attack configurations are necessary to stress-test safety and which are redundant. Answering this would enable more rigorous evaluation with limited computational resources. Future work on the standardized assessment of a large number of attack and defense configurations could reveal crucial patterns to guide the development of future safety approaches.

**4.3.4. How can we scalably evaluate thousands of models?** A major challenge to better understanding the open-weight ecosystem stems from the sheer number of existing models. Coordinated efforts to evaluate their safety properties at scale could improve practical risk management and future risk modeling. For example, platforms like Hugging Face which host and distribute large numbers of AI models can struggle to reliably identify and remove ones that violate their content policies (e.g., Maiberg, 2025). However, ecosystem-level evaluation is complicated by scale, architectural diversity, and the continuous introduction of new models. Evaluations involving tampering attacks can be particularly challenging due to the computational costs of fine-tuning and other tampering algorithms. There is a need for infrastructure for evaluating models at scale that balances efficiency with thoroughness. These approaches might also integrate new technical resources like model provenance techniques (see Section 4.5).

## 4.4 Staged Deployment Strategies

**Gradually increasing access to a model before a full open release helps developers monitor for risks and adjust their safeguards and deployment strategies.** Release strategies for AI systems do not fall into a binary between fully closed and fully open. Different strategies can strike different tradeoffs between open access and centralized control. For example, *beta testing* and *gated access* methods allow developers to make a model available only to a relatively small set of people before it is made fully open (Solaiman et al., 2025). Deploying models in stages can allow developers to gradually increase access while monitoring for unexpected uses and conducting research on potential harms. This allows a developer to refine their approach to safeguards and release before the model is fully open. This section focuses specifically on technical strategies that can serve as intermediate steps in staged deployments.

**Existing work:** There is a spectrum of deployment strategies between fully closed and fully open (Solaiman, 2023). Some of which, such as beta testing, do not have open technical problems related to open-weight

models and are thus out of scope of this paper (see Section 3).[4] However, here we consider several technical strategies on the openness spectrum.

First, *split deployment* strategies divide the model between client devices and server devices. Currently, most research on split learning and inference focuses on either enabling the use of large models on small devices (Xie et al., 2025; Lin et al., 2024; Ren et al., 2023) or keeping user inputs private from developers (Yao & Li, 2024; Mai et al., 2024; Shu et al., 2025).

Second, there more niche technical strategies for restricted forms of deployment that involve *hardware locking* (Clifford et al., 2025) or *homomorphic encryption* (Podschwadt et al., 2022).[5]

**4.4.1. What exfiltration risks do split deployment strategies pose, and how can they be mitigated?** Successful split deployments require that private model layers are kept secure. However, attackers can aim to exfiltrate them using reconstruction (e.g., Shu et al., 2025; Nevo et al., 2024; Carlini et al., 2024) or distillation (e.g., Huangpu & Gao, 2024) methods. Prior work on security focuses on securing client inputs against reconstruction attacks on small models Zhu et al. (2025b); Shu et al. (2025); Shabbir et al. (2025). In this regime, attackers have the upper hand – easily being able to reconstruct small, private portions of a model. In general, it is not well understood how the scale of the model and the proportion of parameters hidden change the cost of effectively reconstructing or distilling private layers of a model. Initial work on adapting this work to LLMs and frontier models broadly Shu et al. (2025); Yao & Li (2024) has also not measured how split location impacts vulnerability to attacks, nor tested the efficacy of a wide variety of attacks. It is not currently well understood how vulnerable private layers of a model are to exfiltration as a function of the architecture, hidden layers, model size, and attack algorithm.

**4.4.2. How can we design split learning and inference APIs that are less costly and more competitive?** A second challenge for split deployment strategies is the induced latency due to communication across the split. The necessity of overhead represents a fundamental limitation compared to other forms of deployment which can make split strategies less competitive. Efficiency challenges are especially acute for autoregressive and diffusion models, which require communication between the server and client for every iteration. This invites future work to develop competitive alternative models and methods Sahoo et al. (2024); Xie et al. (2025); Shen et al. (2025) that reduce the amount of information shared between the server and client, the number of messages passed, and/or the delay from overhead.

**4.4.3. Can hardware locking or homomorphic encryption offer practical options for staged deployment?** First, *hardware locking*, involves linking a model to specific, secure hardware (Clifford et al., 2025). This process certifies a model as "runnable" on a given piece of hardware, creating a secure chain of trust from the hardware to the model itself. Hardware-locking is precedented in traditional software, where hardware security enforcement is used to operate a zero-trust environment. However, designing and deploying infrastructure in an ever-changing open-weight model ecosystem would be challenging. The requirement for highly specialized and secure hardware poses a significant barrier to practical usage, and may be prohibitive for less-resourced developers. It is currently unclear if and how hardware locking could be a useful strategy for staged deployments. Meanwhile, at best, it could only be helpful for safety in niche applications. A second partially open strategy could involve a developer releasing a cryptographically encrypted model and retaining the exclusive ability to homomorphically encrypt inputs for it via an API. Current uses of these techniques focus on privacy-preserving machine learning rather than open-weight deployment (Lee et al., 2022; Podschwadt et al., 2022; Brand & Pradel, 2023; Ebel et al., 2025; Cheng et al., 2025b). However, scale poses a key practical challenge. Existing frameworks can only practically handle models with tens of millions

---

[4]Another such strategy is to use fine-tuning APIs which allow for users to experiment with fine-tuning a closed-weight model (Wu et al., 2024). There are open technical problems related to the safety of fine-tuning APIs such as reliably detecting adversarial attempts at obfuscating harmful fine-tuning data (Halawi et al., 2024; Davies et al., 2025) (see also Section 4.2 and Section 4.3). However, when using a fine-tuning API as a step for staged deployment, a model developer will typically not want to restrict fine-tuning data in order to monitor more realistic misuses of the model when it is deployed with open weights. Thus, malicious users would have little to no incentive to obfuscating harmful fine-tuning data. As such, these challenges are out of scope for this paper (see Section 3).

[5]Deployments involving split inference, hardware locking, or homomorphic encryption do not constitute "open-weight" releases in the traditional sense, as users cannot independently run the full model. We base our discussion of these strategies in this section on the premise that they can serve as intermediate steps in staged deployments that enable monitoring and risk assessment before a model's full open release.

of parameters (e.g., Ebel et al., 2025). It is not clear if homomorphic encryption offers a practical option for frontier models. Like hardware locking, it could only be useful in niche applications.

### 4.5 Model Provenance and Forensics

**Model provenance methods help stakeholders study the spread and uses of open-weight models.** While not directly upstream of model releases, ecosystem monitoring methods are a key component of risk management because they help stakeholders better study the real-world uses and impacts of models. Model provenance and forensics in the open-weight AI ecosystem are key to answering questions such as "What model is this?" and "What modifications has it undergone since its original release?"

**Existing work:** Here, we discuss three complementary types of methods: model watermarking, model heritage inference, and proof of training.

First, *model watermarking* methods aid in the identification of models. In contrast to data watermarking methods (Zhao et al., 2025b), model watermarks refer to model properties that serve to uniquely identify a model or a single instance of a model. Some approaches for model watermarking embed signals during generation without modifying the model's weights, but they depend on specialized decoding algorithms that can be disabled by users (Kirchenbauer et al., 2024a;b). Less-tamperable model watermarking methods must be 'baked into' a model's weights. For example, some methods allow for detection by implanting unique model *behaviors* (e.g., Yu et al., 2021; Fernandez et al., 2023; Xu et al., 2024; Christ et al., 2024). Other model watermarks allow for detection by analyzing model *parameters* by adding noise signatures across the model (e.g., Pagnotta et al., 2024; Block et al., 2025). Additional approaches have also been developed for quantization-based schemes (Li et al., 2023). However, surveys have emphasized ongoing challenges in balancing robustness and imperceptibility tradeoffs (Liang et al., 2024; Boenisch, 2021; Liu et al., 2024a).

Second, in contrast to watermarks, *model heritage inference* methods help researchers study the spread of models in the wild. These techniques have been used to reconstruct a genealogy from weights alone (Horwitz et al., 2025b). Zhu et al. (2025a) and Nikolic et al. (2025) also developed statistical tests determine if two models were trained independently or not. These techniques could offer useful tools to study real-world impacts and enforce licenses.

Finally, *proof of training* methods can be used for verifying that AI systems have undergone training processes with specific properties. Due to the complex supply chains behind some open-weight models, proof of training methods can uniquely enable trust throughout a model supply chain (Jia et al., 2021). Here, recent literature suggested ways to enable provable yet private training provenance using cryptographic methods to create verifiable, records of a model's origins (Garg et al., 2023; Abbaszadeh et al., 2024; Meiklejohn et al., 2025). These methods allow a model developer to verify that their model was trained according to a specific process. For example, they could prove they excluded a certain type of harmful data from their training corpus or applied a specific safety fine-tuning algorithm.

**4.5.1. How can we watermark models in ways that are more durable against common modifications without side effects?** Evaluations suggest that current content attribution watermarks can become undetectable under common open-weight model modifications, such as quantization, fine-tuning, model merging, and pruning (Gloaguen et al., 2025). There is also an absence of standardized benchmarks for comparing watermark durability across realistic combinations of model modifications, such as quantization followed by fine-tuning or merging followed by distillation (Li et al., 2023; Lv et al., 2023). Improving durability requires addressing tensions between watermark subtlety, persistence, and robustness. While some techniques (e.g., Pagnotta et al., 2024) demonstrate significant robustness to removal techniques, they remain vulnerable to distillation or sophisticated tampering attacks combining multiple modification types (Christ et al., 2024). The community has yet to see content watermarks designed for deployments where white-box access for fine-tuning, distillation, model merging, and quantization in sequence are standard practice. This highlights the additional downstream challenge of incorporating these methods into usage frameworks that take their rate of false positives and false negatives into account.

**4.5.2. What algorithms can enable scalable and versatile model heritage inference?** Ecosystem-wide heritage inference is desirable (Horwitz et al., 2025a) but not tractable with current infrastructure and

methods. For example, using current methods (Horwitz et al., 2025b), charting models across a platform such as Hugging Face would require millions of pairwise comparisons between models. While independence between two specific models is computationally inexpensive (Zhu et al., 2025a), continuous ecosystem-wide monitoring must accommodate daily uploads of potentially thousands of new models. Current methods also face four critical limitations beyond computational scaling when simply comparing two models. First, mixed heritage models created through weight averaging, model merging, or 'model soups' remain unaddressed despite growing prevalence. Existing approaches have focused primarily on finetuning and single-parent lineages. Second, cross-architecture techniques to account for processes such as knowledge distillation are neglected. While Zhu et al. (2025a)'s unconstrained setting enables some comparisons through proxy models, systematic handling of diverse architectures demands architecture-agnostic methods. Concurrently with this work, (Kuditipudi et al., 2025) has introduced a black-box heritage inference method with the potential to address this challenge. Third, accurately quantifying the degree of contribution from multiple different parent models requires granular attribution methods. Fourth, adversarial scenarios where actors actively obscure provenance would require more robust methods. A final challenge will be the implementation of heritage inference in ways that are efficient and acccount for their false positives and negatives.

**4.5.3. How practical and scalable are proof of training methods?** Present techniques for proof of training have limitations (Choi et al., 2023; Sun et al., 2025). A major barrier to the practical adoption of provable provenance is computational overhead. Generating zero-knowledge proofs for training runs that involve trillions of datapoints and billions of model parameters is currently computationally prohibitive. The generated proofs must also be integrated into the broader AI ecosystem. This involves creating infrastructure for issuing, storing, and verifying cryptographic certificates. Finally, current research primarily focuses on verifying straightforward properties of the training process, such as the inclusion or exclusion of specific data. However, many advanced safety techniques involve nuanced procedures that are difficult to formalize and verify cryptographically. An open challenge is to extend proof of training methods to cover more qualitative safety-related interventions. However, at best, even if proof of training methods can be practically scaled and implemented, they could only be useful for safety in niche applications.

# 5 What techniques are prominent open-weight developers reporting on?

To understand what frontier open-weight model developers have reported about technical safeguards, we analyzed technical reports and model cards from popular open-weight models. We selected two sets of models. First, we identified the 10 most widely adopted open-weight models on Hugging Face. We selected these models by examining Hugging Face download statistics[6] as of Oct 15 2025: specifically, we selected the top 10 organizations by total model downloads (all time) that released foundation models[7] in 2025, then identified the most downloaded model from each organization. For model families released under shared documentation (e.g., Qwen3-0.6B, Qwen3-4B, Qwen3-8B), we report on the model family as a whole. This yielded the following models: Qwen3 (Yang et al., 2025), DeepSeek-R1 (Guo et al., 2025), Gemma3 (Team et al., 2025a), gpt-oss (Agarwal et al., 2025), Nemotron-Nano (Basant et al., 2025), Granite-3.3 (Granite Team, 2024), Phi-4 (Abdin et al., 2024), EXAONE-Deep (LG et al., 2025), Llada-8B (Nie et al., 2025), and GLM-4.5Team et al. (2025b).

Second, we examined specific image and video generation models that Hawkins et al. (2025) and Kamachee et al. (2025) highlighted as being commonly used for image and video deepfakes. These models included: Stable Diffusion 1.x (Rombach et al., 2021), FLUX (Batifol et al., 2025),[8] Wan2.x (Wan et al., 2025), HunyuanVideo (Kong et al., 2025), and LTXV (HaCohen et al., 2024).

While not exhaustive, these models represent both highly-adopted releases and models with documented misuse patterns, spanning multiple organizations, jurisdictions, architectures, and modalities.

---

[6] https://huggingface.co/spaces/evijit/ModelVerse

[7] We specifically look at models that are supported by the `text-generation` pipeline of the `transformer` library, as these constitute the vast majority of foundation models in popular use. Some of these models natively support multimodality.

[8] Batifol et al. (2025) postdates Hawkins et al. (2025), but Batifol et al. (2025) is the only technical report available for any FLUX model, so we analyze it in the Table 2.

| Model | Organization | Safe Data Curation (Section 4.1) | Tamper-Resistance Training (Section 4.2) | Tampering Evals (Section 4.3) | Staged Deployment (Section 4.4) | Model Provenance (Section 4.5) |
|---|---|---|---|---|---|---|
| *Most-Downloaded LLMs/Multimodal Foundation Models on Hugging Face that were released in 2025* | | | | | | |
| **Qwen3** (Yang et al., 2025) | Alibaba | 1-3 Sentences | No Mention | No Mention | No Mention | No Mention |
| **DeepSeek-R1** (Guo et al., 2025) | DeepSeek | No Mention | No Mention | No Mention | No Mention | No Mention |
| **Gemma3** (Team et al., 2025a) | Google | 1-3 Sentences | No Mention | No Mention | No Mention | No Mention |
| **gpt-oss** (Agarwal et al., 2025) | OpenAI | Paragraph | No Mention | Dedicated Paper | No Mention | No Mention |
| **Nemotron-Nano** (Basant et al., 2025) | NVIDIA | Paragraph | No Mention | No Mention | No Mention | No Mention |
| **Granite-3.3** (Granite Team, 2024) | IBM | Dedicated Section | No Mention | No Mention | No Mention | No Mention |
| **Phi-4** (Abdin et al., 2024) | Microsoft | Paragraph | No Mention | No Mention | No Mention | No Mention |
| **EXAONE-Deep** (LG et al., 2025) | LG AI Research | No Mention | No Mention | No Mention | No Mention | No Mention |
| **Llada-8B** (Nie et al., 2025) | GSAI-ML | No Mention | No Mention | No Mention | No Mention | No Mention |
| **GLM-4.5** (Team et al., 2025b) | Z.AI | No Mention | No Mention | No Mention | No Mention | No Mention |
| *Models highlighted in Hawkins et al. (2025) and Kamachee et al. (2025)* | | | | | | |
| **Stable Diffusion 1.x** (Rombach et al., 2021) | Stability AI | No Mention | No Mention | No Mention | No Mention | No Mention |
| **FLUX** (Batifol et al., 2025) | Black Forest Labs | 1-3 Sentences | No Mention | No Mention | No Mention | No Mention |
| **Wan2.x** (Wan et al., 2025) | Alibaba | 1-3 Sentences | No Mention | No Mention | No Mention | No Mention |
| **Stable Video Diffusion** (Blattmann et al., 2023) | Stability AI | No Mention | No Mention | No Mention | No Mention | No Mention |
| **HunyuanVideo** (Kong et al., 2025) | Tencent | No Mention | No Mention | No Mention | No Mention | No Mention |
| **LTX-Video** (HaCohen et al., 2024) | Lightricks | No Mention | No Mention | No Mention | No Mention | No Mention |

Table 2: **What technical safety techniques are prominent open-weight model developers reporting on?** We overview what open-weight model risk management techniques are discussed in technical reports. The top section includes the top 10 most-downloaded language/multimodal model families released between January and October 2025 (by organization). The bottom section includes image and video generation models highlighted by Hawkins et al. (2025) and Kamachee et al. (2025) as being prominently used for image and video deepfakes. This table offers a source of reference documenting developer disclosures. This table does not analyze substance and is not intended to be a scorecard. *Legend:* NM No Mention, 1-3S 1-3 Sentences, P Paragraph, DS Dedicated Section/Paper. "No mention" does not imply no implementation. We only focus on safety – for example, we do not analyze reporting on quality-focused data curation.

**Summarizing reporting on technical safeguards:** For each of the five categories of safeguards discussed in Section 3, we examined whether each model's documentation reported on the use of these techniques for improving safety. We categorized reporting as: **no mention**, a **1-3 sentence mention**, a **paragraph-level description**, or a **dedicated section/paper**. We note that this table is qualitative in nature and is meant to analyze overall safety reporting trends by open model developers without necessarily highlighting any particular organization or model. "No mention" does not imply "not implemented", and our analysis does not consider the substance or effectiveness of reported techniques. Our observations are shown in Table 2.

Our analysis reveals several patterns in how open-weight model developers report on technical safeguards. Among the top open model developers by downloads of models released in 2025, data curation is the most commonly reported safeguard, with 6 out of 10 models providing at least brief documentation—ranging from 1-3 sentences (Qwen3, Gemma3) to dedicated sections (Granite-3.3). Three models provide paragraph-level descriptions (gpt-oss, Nemotron-Nano, Phi-4), though notably some of these focus on post-training or mid-training safety fine-tuning rather than pre-training data filtering specifically which may be key for tamper-resistent safeguards (see Section 4.2). However, documentation for other technical open-weight

model safeguards remains sparse. Tamper-resistant training algorithms recieve no mention in any of the analyzed models. Tampering evaluations appear in only one model (gpt-oss), which dedicated a separate paper to adversarial fine-tuning assessments (Wallace et al., 2025). Staged deployment strategies and model provenance/forensics techniques are absent from all technical reports examined.[9]

Among the five image and video generation models highlighted in Hawkins et al. (2025) and (Kamachee et al., 2025), similarly few safeguards are reported across models with only the FLUX and Wan2.x technical reports making mention of safety-focused data curation (Batifol et al., 2025; Wan et al., 2025).

**Implications:** The prevalence of grey cells in Table 2 suggests substantial room for growth in the science of technical open-weight model safety. The general absence of reporting on tamper-resistance, tampering evaluations, and provenance techniques is particularly notable given the vulnerability of open-weight models to tampering attacks. This gap between documented risks and reported mitigations suggests that either: (1) these techniques are not being widely implemented, (2) they are being implemented but not documented, or (3) effective methods for these safeguards remain underdeveloped.

These findings align with our broader argument that building the science of open-weight model risk management requires not only developing new technical safeguards, but also establishing norms around transparent reporting of safety practices. The scarcity of documentation across multiple safeguard categories suggests the field could benefit from more thorough reporting on technical risk mitigation strategies.

# 6 Discussion

**Significance:** Increasingly capable open-weight models are being released on a regular basis, with research showing open-weight model capabilities to consistently be only 6-12 months behind frontier proprietary models (Cottier, 2024; Maslej et al., 2025). There are clear benefits of open-weight model development. These include driving innovation, enabling AI safety/security research, enabling flexible AI adoption, and spreading benefits and access to AI (Seger & Hancock, 2025). However, open-weight models also pose distinct risks stemming from the potential for rapid proliferation of model flaws and the ease with which malicious actors can bypass safeguards against misuse. We believe that a positive future with AI will involve a balance of proprietary and open-weight model development. Effective tools to mitigate risks will not only be key for mitigating open-weight models' risks but also accessing their benefits by avoiding backlash (Henderson et al., 2023). Toward this end, this paper investigates technical interventions that could help mitigate and monitor risks from open-weight AI models. Our collective hope is that this paper will help to build the field of technical open-weight model risk management.

**Limitations:** As discussed in Section 3, this paper only focuses on technical tools with distinct implications for open-weight models. This focus is not meant to imply that open problems, strategies distinct to open-weight models, or technical strategies are the most useful or important. We concur with François et al. (2025), Srikumar et al. (2024), and UK AISI (2025) that a holistic approach to monitoring and mitigating risks in the open-weight model ecosystem will be crucial. However, not all techniques for open-weight model safety will be equally effective or competitive. Research on the limitations and practicality of techniques will be important for refining the toolkit.

**Uncertainties:** It is unclear how effective different safeguards for open-weight models will ultimately be. Not all approaches will be equally effective. It is also unclear how much counterfactual risk open-weight models will pose compared to closed-weight models (Kapoor et al., 2024). Thus, we emphasize the value of gathering more information about open-weight models through additional research and analysis of impacts across the ecosystem. In doing so, the research community should be mindful of both 'openness washing' (Grieve, 2024) and 'safety washing' (Ren et al., 2024). It is important for researchers and policymakers to be open to evidence both in favor of and against the possibility that some models may pose large risks if deployed in certain ways. Some models – even with safeguards – might enable acute misuse if deployed

---

[9]Several organizations in Table 2 have implemented safeguards not specific to open-weight models and/or released companion safety/guardrail models alongside their base models, including Qwen3Guard Zhao et al. (2025a), Granite Guardian Padhi et al. (2024), and NemoGuard Rebedea et al. (2023). While these external safety tools are out of scope for this analysis, they represent meaningful contributions to the open-weight safety ecosystem.

with open weights. Others might significantly hinder open-science or concentrate large amounts of power if deployed with closed weights. Others still might pose major risks regardless of deployment type.

**Incentivizing future research:** While we are optimistic about the potential value of more research into technical mitigations against open-weight model risks, we recognize that incentives for private actors to research and develop robust safeguards for frontier open-weight models are currently limited. Furthermore, technical safeguards for open models will only be resistant to some degree of intervention. So from a researcher's standpoint, work on technical interventions may be high-risk (in terms of investment) and limited reward. This does not mean, however, that this work is not worthwhile. Each of the strategies we discussed in Section 4 is individually imperfect, but contributes meaningfully to reducing harm or increasing information. Used in concert, these methods can substantially improve risk management. There are also barriers to important safety and security research that remain in place. While many open developers provide models with the intent of enabling researcher, and participate in open flaw bounties (McGregor et al., 2025), some major open-weight developers do not consistently offer legal 'safe harbors' and even impose legal language or technical obstacles against good-faith safety evaluations into their systems' safeguards (Longpre et al., 2024a).

**The importance of openness (not just of model weights):** The status quo may currently incentivize little openness related to open-weight model risk management (Table 2). However, in building the science of open-weight model risk management, we emphasize the value of open scientific collaboration (Phang et al., 2022; Linåker et al., 2025; Scotti, 2025), open research (Biderman et al., 2023; Liu et al., 2023; Groeneveld et al., 2024), open evaluations (Gao et al., 2021; Bommasani et al., 2023; Biderman et al., 2024), open reporting about risk-management methodology (Seger, 2024), and open standardized documentation. Just as building the science of open-weight model risk management will provide a collective good, it will also require collective effort.

## Acknowledgments

We are thankful to Anka Reuel, Isabella Duan, Jack Sanderson, Nicholas Carlini, and Stella Biderman for discussions on drafts of the paper.

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
