# OpenReview forum: "Open Technical Problems in Open-Weight AI Model Risk Management"
_TMLR — Accepted by TMLR_

### Review · Reviewer_hFix · 2025-11-22

**Summary Of Contributions:**

The authors present a valuable survey and taxonomy regarding the unique risk management challenges of open-weight AI models. Unlike API-based models, open-weight models allow users to bypass standard safeguards, necessitating different technical approaches. The paper outlines 16 open technical problems across five lifecycle stages, including training data curation, tamper-resistant training, and ecosystem monitoring. Additionally, the authors conduct a useful gap analysis of technical reports from prominent models (e.g., Qwen3, gpt-oss), demonstrating that safeguards like tamper-resistance and provenance are currently under-reported by developers

**Audience:**

Yes

**Audience Explanation:**

This work can be relevant to TMLR's audience, particularly researchers in AI safety, policy, and open-source development. The structured taxonomy provides a helpful roadmap for future research , and the consolidated analysis of current developer reporting practices offers a useful reference point for the community.

**Broader Impact Concerns:**

The paper focuses on risk mitigation and does not present direct ethical concerns.

**Claims And Evidence:**

Yes

**Claims Explanation:**

The paper provides a well-supported characterization of the "current state of the field". The authors back their claims with a comprehensive literature review and an empirical audit of 15 recent frontier model technical reports in Section 5 . The distinction drawn between the risk surface of closed models versus open-weight models is logically sound and accurately reflects current technical realities.

**Requested Changes:**

- Theoretical Limits of Tamper Resistance (Strengthen): In Section 4.2, please explicitly discuss the theoretical difficulty (or potential impossibility) of achieving robust "tamper resistance"  when adversaries have white-box access to weights and gradients. Acknowledging these fundamental limits would strengthen the technical grounding of the proposed solutions.

- Feasibility of Hardware Locking: The discussion on hardware locking in Section 4.4.3 would benefit from a stronger critique regarding its practicality. As noted, specialized hardware requirements pose significant barriers , and expanding on the friction this creates for the open ecosystem would add necessary nuance.

- Table 2 Clarification: Please clarify in Section 5 whether "No Mention" allows for the possibility that developers perform these evaluations (e.g., internal red teaming) but choose not to disclose them to avoid providing attack roadmaps. This can be an interesting avenue as well.

---

> ### Author Response · Authors · 2025-11-27
> **Thanks + response**
>
> We are thankful for your time and help, especially related to the limits of techniques. We were glad to hear that you found the paper to be a helpful roadmap for future research! Here are our replies to the updates you discussed.
>
> ## 1. Limits of tamper-resistance
>
> > explicitly discuss the theoretical difficulty (or potential impossibility) of achieving robust "tamper resistance" when adversaries have white-box access
>
> **Action 1.1:** Thanks, we added to section 2 to say how full tamper robustness is impossible and the practical challenge is improving the degree of tamper resistance. We also added to section 4.3.2 about how and why we believe that dataset construction is the most key bottleneck for making malicious fine-tuning difficult.
>
> ## 2. Feasibility of Hardware Locking
>
> **We agree.** We generally believe that hardware locking is likely to be lower priority than most other open problems we discuss.
>
> **Action 2.1:** We are adding to section 4.4 to explicitly discuss how hardware locking, at best, could only be useful for niche applications.
>
> ## 3. Does “no mention” imply “not done”?
>
> > Please clarify in Section 5 whether "No Mention" allows for the possibility that developers perform these evaluations (e.g., internal red teaming) but choose not to disclose them
>
> **Action 3.1:** Yes, this is entirely possible. We added to section 5 and the table caption to explicitly say that no mention does not imply that something was not implemented. See also response point 3 from Reviewer bDvJ.
>
> –
>
> Thanks again! We are curious about your thoughts.

---

> > ### Comment · Reviewer_hFix · 2025-12-02
> >
> > Thank you for the changes. It answers most of my concerns.

---

### Review · Reviewer_mc7U · 2025-11-23

**Summary Of Contributions:**

This paper identifies several open technical problems related to the risks posed by open-weight frontier AI models, structured across the following lifecycle stages: training data curation, tamper-resistant training & unlearning algorithms, tampering-aware evaluations, staged deployment methods, as well as model provenance and forensics.

This submission’s motivation is that open-weight models have unique safety challenges relative to closed models: they can be modified freely, disseminated irreversibly, and used without oversight. The authors survey existing work, identify gaps, and articulate open research problems. The paper concludes with a small empirical analysis of the safety-related reporting practices of current open-weight model developers.

**Audience:**

Yes

**Audience Explanation:**

This submission contributes a concrete set of open problems, many of which are indeed pressing and under-studied. These include making data filtering robust across harm categories, building tamper-resistant unlearning, benchmarking tampering attacks, watermarking or tracking modified models in the wild, evaluating thousands of ecosystem models efficiently. These iarevaluable to guide research priorities.

The submission also provide novel and interesting empirical analysis. The study of reporting practices across highly downloaded 2025 open-weight models (e.g., Qwen3, Gemma3, gpt-oss) is informative and highlights striking gaps — especially the lack of reporting on tamper resistance, tampering evaluations, staged deployment, and provenance.

**Claims And Evidence:**

Yes

**Claims Explanation:**

This submission has clear and timely motivation. The introduction clearly establishes the escalating capabilities of open-weight models and the mounting difficulty of governing them due to unrestricted access and rapid proliferation. This framing is compelling and grounded in evidence from recent technical reports and open-weight releases. The paper has well-structured taxonomy. The five-part organization (training data → training → evaluations → deployment → monitoring) gives the reader a coherent conceptual map of what “open-weight model risk management” consists of. This structure mirrors and synthesizes multiple taxonomies in prior work. This submission also preent excellent survey of existing work. The authors assemble a broad range of relevant research and surface nuanced issues (e.g., the difficulty of filtering multilingual data; limits of post-training tamper resistance; exfiltration risks in split inference). The survey is one of the strongest components.

**Requested Changes:**

This paper lack of prioritization for all the open problems. All problems are presented with roughly equal weight, but some are clearly more urgent or tractable than others. A prioritization framework—based on risk severity, tractability, or expected impact—would increase the paper’s utility for guiding the field.

There are also missing discussion of failure modes in “technical safeguards” themselves. Some safeguards (watermarking, unlearning, tamper resistance) may introduce new vulnerabilities like false negatives in forensics, over-confident safety claims leading to risk compensation, backdoors introduced during watermarking, and reduction in benign capabilities due to unlearning. These tradeoffs are only partially acknowledged. For each safeguard category, discuss risks introduced by the safeguard itself and highlight where these create second-order hazards.

The paper sometimes mixes “unique to open-weight” vs. “more challenging for open-weight”. The authors intend to focus on strategies with unique implications for open-weight models. However, tamper-resistant training is relevant for API-based fine-tuning in closed models. Heritage inference is useful for closed-model auditing. Split deployment and hardware locking are not “open-weight” strategies but restricted-weight strategies. This definitional ambiguity weakens the conceptual crispness.

The empirical developer-reporting study in this submission lacks clear methodology. The sampling procedure is described but could be more rigorous: How were “downloads” normalized across model families? How were ambiguous documentation statements interpreted? Were multiple reviewers involved to improve reliability?

There are limited discussion of non-technical constraints and incentives. Although the paper notes that incentives for private actors to implement safeguards are weak, the implications of this are under-explored. Many proposed safeguards (e.g., proof-of-training, hardware locking) are unlikely to be adopted without external pressure. The authors should expand the discussion about incentive structures, deployment economics and incentive-aligned governance mechanisms.

---

> ### Author Response · Authors · 2025-11-27
> **Thanks + response**
>
> We are thankful for your time and help, especially related to prioritization and drawbacks. We were glad to hear that you found the paper to be timely and helpful! Here are our replies to the updates you discussed.
>
> ## 1. Prioritization framework
>
> > A prioritization framework—based on risk severity, tractability, or expected impact—would increase the paper’s utility for guiding the field.
>
> **First, we agree.** We think that one of the key ways a paper like this can provide value is by helping readers distinguish what is more versus less promising.
>
> **On the other hand, we think that a formal framework for prioritization that reflects empirical findings and consensus would be hard.** Priority is a function of both potential value and tractability, but tractability is inherently hard to determine for open problems. Instead, we think that discussing considerations related to prioritization throughout the paper is a more flexible approach.
>
> **Action 1.1:** We are adding discussions of:
> * How hardware locking, homomorphic encryption, and proof of training methods could, at best, only be useful for niche applications. (Sections 4.4 and 4.5)
> * How determining the relative promisingness of different methods is itself an open problem, and not all approaches will be equally promising (Section 6).
>
> **Question 1.2:** Are there other considerations you would recommend adding?
>
> ## 2. Discussing drawbacks
>
> > Some safeguards (watermarking, unlearning, tamper resistance) may introduce new vulnerabilities…
>
> **Action 2.1:** We revisited and emphasized the following to boost discussion of drawbacks:
> * False negatives in forensics, including both watermarks and model heritage inference (Section 4.5)
> * Off-target effects of data filtering, unlearning, or watermarking (Sections 4.1, 4.2, and 4.5). We now cite [Zhou et al. (2024)](https://arxiv.org/abs/2408.00376).
> * Expanding our discussion of reduction in benign capabilities from data filtering and unlearning (Sections 4.1 and 4.2)
> * Further discussing overhead induced by split deployment APIs (Section 4.4)
>
> **Question 2.2:** Anything we missed? Do you recommend any particular papers for us to make sure we discuss?
>
> ## 3. Unique to open-weight vs more challenging for open-weight
>
> > The paper sometimes mixes “unique to open-weight” vs. “more challenging for open-weight”.
>
> **We see what you mean, and it is worth being clearer about this.** Our intended criterion is to include techniques that have *some* distinct implications/applicability for open-weight models, regardless of how similar these are to implications/applicability for closed-weight models. In other words, we try to cast the net broadly, encompassing what you describe as problems that are “more challenging for open-weight” models.
>
> **Drawing a crystal clear distinction is a bit difficult.** The biggest challenges, like we frame here, often have varied implications. Also, there is not an open/closed binary distinction between model deployments. As you mention, closed-weight fine-tuning APIs are an example.
>
> **Question 3.1:** Do you have suggestions about how we might change our criterion and footnote 3? Should we say “acute” or “distinct” instead of “unique”? Should we say “relevance” instead of “applicability”
>
> ## 4. Developer reporting questions
>
> > How were “downloads” normalized across model families?
>
> We aggregated download counts across model families with no discounting or normalization related to when in 2025 they were released.
>
> > How were ambiguous documentation statements interpreted?
>
> Empirically, we believe we only encountered one genuine ambiguity. It related to whether to count training data curation efforts if they were only used during fine-tuning. However, **we always erred on the side of being more generous to the developers.**
>
> > Were multiple reviewers involved?
>
> Two of us split the effort and asked each other to review potential ambiguities, which ended up being the above.
>
> ## 5. Discussing incentives
>
> > The authors should expand the discussion about incentive structures, deployment economics, and incentive-aligned governance mechanisms.
>
> **Pushing back slightly.** We agree with the relevance of these topics. But we also believe that an in-depth discussion of incentives and how to shape them merits separate work.
>
> **Action 5.1:** We added to the last two paragraphs of section 6 a mention of how status quo incentives may lead to little transparency, along with a pointer to Table 2.
>
> **Question 5.2:** Any papers you recommend we make sure to cite related to this?
>
> –
>
> Thanks again! We are curious about your thoughts.

---

> > ### Comment · Reviewer_mc7U · 2025-12-08
> >
> > Thanks for the rebuttal, this rebuttal solves all of my concerns

---

### Review · Reviewer_bDvJ · 2025-11-26

**Summary Of Contributions:**

This paper provides a structured overview of the open technical challenges in managing risks associated with open-weight AI models. The authors categorize sixteen open problems into five stages of the model lifecycle: data curation, tamper-resistant training, tampering evaluations, staged deployment, and model provenance/forensics—and analyze how current open-weight model reports address (or fail to address) these issues. The work aims to establish a clearer research agenda and highlight gaps in current practices, especially the lack of standardized evaluations for post-release model tampering.

Key strengths:
1. Provides a clear, systematic taxonomy of risk related technical challenges unique to open-weight settings.
2. Offers practical guidance on where current open-weight reports fall short, supported by a cross-model qualitative analysis.
3. The survey of technical reports from major 2025 releases (Table 2) provides timely and concrete evidence of the disconnect between theoretical safety techniques and actual industry practices.

Key weaknesses:
1. The model set analyzed in Section 5 is small and focused on high-profile systems, which may introduce sampling bias.
2. The paper functions primarily as a position/survey piece: it organizes challenges and open problems but does not contribute new algorithms, proofs, or experimental methods.
3. The Section 5 analysis depends solely on developer-provided reports rather than independent verification. “No Mention” does not imply the absence of a mitigation, reducing the strength of the survey’s conclusions.

**Audience:**

Yes

**Audience Explanation:**

The paper addresses topics, such as open-weight model safety, tamper-resistant training, and provenance mechanisms, that are directly relevant to researchers working on foundation models, security, and responsible AI. Its taxonomy of open technical problems provides a structured perspective that many in the TMLR community may find useful for guiding future research directions. Given the growing importance of open-weight models, these findings would likely interest a meaningful subset of TMLR’s audience.

**Broader Impact Concerns:**

The work focuses on risk mitigation, presenting low direct ethical risk. However, a Broader Impact Statement should acknowledge a mild dual-use concern: the detailed discussion of tampering evaluations and attack vectors could theoretically assist adversarial actors in bypassing safeguards.

**Claims And Evidence:**

Yes

**Claims Explanation:**

The paper’s main claims are supported by clear reasoning and well cited prior work. Its conceptual taxonomy is logically developed, and the identified technical gaps are grounded in existing literature. The Section 5 analysis of 2025 technical reports provides relevant evidence for the claim that current safety reporting is inadequate. However, this empirical component relies solely on developer-provided documentation rather than independent verification, which weakens the evidential strength. Overall, the claims are appropriately supported for a position paper, though some conclusions would benefit from stronger methodological rigor.

**Requested Changes:**

Strengthening:
1. Expand Model Survey: The survey focuses on 10 high-profile models. Expanding the dataset or at least discussing why other categories (e.g., smaller open-weight models, community finetunes, multilingual models) were not included would strengthen the representativeness of the ecosystem wide claims.

2. Propose Concrete Metrics: For several open technical problems (e.g., tamper resistance, unlearning stability), the paper would be strengthened by proposing possible evaluation metrics or standardized protocols—such as “steps-to-harmfulness” benchmarks for adversarial fine-tuning—rather than only describing challenges qualitatively.

---

> ### Author Response · Authors · 2025-11-27
> **Thanks + response**
>
> We are thankful for your time and help, especially related to progress measures and making sure that section 5 does its job well. We were glad to hear that you believe that the TMLR community can benefit from the paper.
>
> ## 1. Section 5 only studies 16 systems
>
> > The model set analyzed in Section 5 is small
>
> **In defense of our approach:** We believe that it is useful to center our focus on recent models and popular models. These models tend to come from larger developers who have the greatest ability to implement and report on safeguards. Our 16 models span text, image, video, and multimodal models. We also believe that looking at 16 systems is enough to illustrate the central finding -- that there is not consistent reporting about if and how mitigations are implemented.
>
> **Question 1.1:** Do you have any thoughts on how to best expand the analysis? We are happy to add more systems. By default, we might do this by lowering the popularity threshold. But do you have any ideas about how and how much to expand?
>
> ## 2. The paper is a survey
>
> > The paper functions primarily as a position/survey piece
>
> **No disagreement here.** However, we would emphasize that we wrote this paper to help build the science and enable more technical work. One thing that makes this paper different from many survey works is that it does not focus on documenting the existing state of a field, but entirely on how to build a nascent one.
>
> **TMLR regularly accepts survey works** that reviewers believe are well-constructed and of community interest/value.
>
> ## 3. “No mention” does not imply no implementation
>
> **We agree. The focus on reporting – not implementation – was our goal.** Our goal with section 5 was not to imply that these developers are not implementing safeguards or that their models are not safe. We do not make either of these claims. Instead, our goal was to highlight the transparency about if/how safeguards were implemented because of transparency’s role in building the science. In section 6, we discuss how *“In building the science of open-weight model risk management, we emphasize the value of open scientific collaboration, open research, open evaluations, open reporting about risk-management methodology, and open standardized documentation.”* From this perspective, transparency about if/how safeguards were used rather than the implementation of those safeguards is the key.
>
> **Action 3.1:** We added that “No mention” does not imply no implementation to the table 2 caption.
>
> **Action 3.2:** We added a reference to table 2 in section 6.
>
> ## 4. Concrete metrics
>
> > For several open technical problems…the paper would be strengthened by proposing possible evaluation metrics
>
> **We already do this somewhat.** For example, we discuss the precision/recall/efficiency frontier for dataset curation and the need for benchmarks for model tampering evaluations.
>
> **Action 4.1:** We have added discussion of progress measures where we have found it applicable:
> * Section 4.2.2: Measures of surgical knowledge editing, including multi-hop reasoning and side effects.
> * Section 4.3.1: Measures for tamper resistance in terms of tokens, steps, compute, and effort.
> * Section 4.4.2: Measures of information flow, number of queries, and inference efficiency for split inference APIs.
> * Section 4.5.2: The precision/recall/efficiency frontier for model heritage inference.
>
> –
>
> Thanks again! We are curious about your thoughts.

---

### Review · Reviewer_Wqgx · 2025-11-28

**Summary Of Contributions:**

This paper discuss open technical problems in open-weight AI model. This issues, in my understanding is the most fundamental problem in AI safety field, and in my understanding, is almost impossible to fully addressed in the current technical context.

**Additional Comments:**

See requested change. I will participate in discussion with constant interation with the authors. I think this paper is important, but at the same time I think several lines of important research directions are missing.  I think I should help to better characterize this paper in these technical aspect.  Feel free to disagree my view in the rebuttal if your feel these lines of research are not relevant and I am open to discuss.

**Audience:**

Yes

**Audience Explanation:**

This survey is important to identify what's the most fundamental problem in safety field. Definitely of interest to the readers.

**Broader Impact Concerns:**

No concerns.

**Claims And Evidence:**

Yes

**Claims Explanation:**

This is a survey paper. Comprehensive reference to prior research has supported the claim of this paper.

**Requested Changes:**

Several research directions on open-weight LLMs are not well discussed.

Section 4.1, when talking about data curation, I suggest the authors to expand discussion on data curation based on different stages. Particularly, data curation for safety has been studied from three main aspects.

* **How to fitler harmful data before the pre-training stage?**

As far as I know two research falls in this angle. The authors should have enough knowledge on this aspect, but should also mention [2].

[1] Deep Ignorance: Filtering Pretraining Data Builds Tamper-Resistant Safeguards into Open-Weight LLMs https://arxiv.org/abs/2508.06601

[2] Best Practices for Biorisk Evaluations on Open-Weight Bio-Foundation Models https://arxiv.org/abs/2510.27629

* **How to contruct better safety alignment data  at the safety alignment stage (after the pre-training) to make the model immune to tampering attack?**

 There are several research along this line, and should be discussed by the authors.  Example reference is include your task may vary[3] and Pharmacist[4].

[3] Why LLM Safety Guardrails Collapse After Fine-tuning: A Similarity Analysis Between Alignment and Fine-tuning Datasets

[4] Pharmacist: Safety Alignment Data Curation for Large Language Models against Harmful Fine-tuning

* **How to construct safety alignment data/filter harmful dta in the tampering attack phase?**

There are a line of research in this line as well and should be discussed. For example. [5][6][7]  explore safety data curation in the tampering attack stage.

[5] Do as I do (Safely): Mitigating Task-Specific Fine-tuning Risks in Large Language Models

[6] When Style Breaks Safety: Defending LLMs Against Superficial Style Alignment

[7] SPARD: Defending Harmful Fine-Tuning Attack via Safety Projection with Relevance–Diversity Data Selection

On the other hand,  [9][10][11][12] [13] explore  harmful data filteration in the tampering attak phase and should be disccussed.

[9] SEAL: Safety-enhanced Aligned LLM Fine-tuning via Bilevel Data Selection (ICLR2025)

[10] Antibody: Strengthening Defense Against Harmful Fine-Tuning for Large Language Models via Attenuating Harmful Gradient Influence https://openreview.net/forum?id=qur2ef8MqQ

[11] Adaptive Defense against Harmful Fine-Tuning via Bayesian Data Scheduler (NeurIPS2025)

[12] Safety-Aligned Weights Are Not Enough: Refusal-Teacher-Guided Finetuning Enhances Safety and Downstream Performance under Harmful Finetuning Attacks https://openreview.net/forum?id=OK2GR1guwv

[13] GradShield: Alignment Preserving Finetuning


Section 4.2,  I think alignment stage defenses against tampering attack (or harmful fine-tuning) can be separated into twp directions.
First line is represented by TAR, which the authors I guess is pretty familiar, but recently there is another line of research which is not discussed by this paper.

* The first line of research is concerning **how to make safety alignment stronger** such that the model cannot learn harmful knowledge, represented by Vaccine, RepNoise, TAR. However, there are some missing references on this line of research, which should be added, i.e., [17-31]

[14] Vaccine: Perturbation-aware alignment for large language model aginst harmful fine-tuning https://arxiv.org/abs/2402.01109

[15] Representation noising effectively prevents harmful fine-tuning on LLMs https://arxiv.org/abs/2405.14577

[16] Tamper-Resistant Safeguards for Open-Weight LLMs https://arxiv.org/abs/2408.00761


**(Missing referece on defesense similar to TAR)**

[17] Booster: Tackling harmful fine-tuning for large language models via attenuating harmful perturbation

 [18]  Leveraging Catastrophic Forgetting to Develop Safe Diffusion Models against Malicious Finetuning

[19] Identifying and Tuning Safety Neurons in Large Language Models

[20] Targeted Vaccine: Safety Alignment for Large Language Models against Harmful Fine-Tuning via Layer-wise Perturbation

[21] Preserving Safety in Fine-Tuned Large Language Models: A Systematic Evaluation and Mitigation Strategy

[22] On Weaponization-Resistant Large Language Models with Prospect Theoretic Alignment

[23] Fight Fire with Fire: Defending Against Malicious RL Fine-Tuning via Reward Neutralization

[24] Model Immunization from a Condition Number Perspective

[25]  Vulnerability-Aware Alignment: Mitigating Uneven Forgetting in Harmful Fine-Tuning

[26]  Locking Open Weight Models with Spectral Deformation

[27] LoX: Low-Rank Extrapolation Robustifies LLM Safety Against Fine-tuning

[28] Towards Resilient Safety-driven Unlearning for Diffusion Models against Downstream Fine-tuning

[29] TOKEN BUNCHER: Shielding LLMs from Harmful Reinforcement Learning Fine-Tuning

[30] AntiDote: Bi-level Adversarial Training for Tamper-Resistant LLMs

[31] Antibody: Strengthening Defense Against Harmful Fine-Tuning for Large Language Models via Attenuating Harmful Gradient Influence

* The second line of research is more interesting. As demonstrated by Xiangyu attack paper, first line of defense, e.g., TAR and RepNoise are still not robust enough, as they are vulnerabe when the fine-tuning learning rate is large. To address this, the second line of  two subsequent research (CTRAP[32] and SEAM[33] ) explore another direction, which is to **embed a collapse trap during safety alignment to make the model collapse if harmful fine-tunig is detected**. I personnally think the collapse trap direction represented by CTRAP and SEAM is the most promising direction to eliminate the fine-tuning risk. Basically, they embed a "shut-down button" into the openweight LLMs and will be activated if the safety alignment is compromised (or the harmful knowledge is actiated) by harmful fine-tuning. I think the authors can expand discussion here.

[32] CTRAP: Embedding Collapse Trap to Safeguard Large Language Models from Harmful Fine-Tuning https://arxiv.org/abs/2505.16559

[33] Self-Destructive Language Model https://arxiv.org/abs/2505.12186

---

> ### Author Response · Authors · 2025-11-30
> **Thanks + response**
>
> We are thankful for your time and help. We are glad that you think the paper is relevant and important. We are thankful to have received this type of review. We are glad that your in-depth notes and familiarity have made the review so directly helpful in improving the thoroughness of the paper. Some of the papers you discussed are from the past month or openreview only, so we are glad for how in-tune you are with the lit. After the TMLR process ends, please feel free to find us and email us if you’d ever like to talk.
>
> ## Additions
>
> Here are the details of where we updated the paper in response to the above.
>
> As you mentioned, a few of the works you included were already ones we discussed: [1], [14-16].
>
> We added one paper to section 4.1: [2].
>
> We added several to section 4.2: [10], [17-20], [22-24], [26-33]
>
> We added several to section 4.3: [3], [5-9], [11-13], [25],
>
> We added one to multiple sections: [4]
>
> And there was one that we did not add because the paper only tests standard training algorithms (SFT, PPO, and DPO) as defenses: [21].
>
> If you would like, please check the specifics of how we incorporated these in the revised draft.
>
> Finally, we have a question: could you tell us a bit about which of these may warrant the most extensive updated discussion in the paper? Specifically, we would value your critical thoughts on which papers give the most useful evidence about what methods research in 2026 should prioritize or investigate.
>
> –
>
> Thanks again! We are curious about your thoughts.

---

> ### Comment · Reviewer_Wqgx · 2025-12-02
> **Thanks for the revision**
>
> Thanks for the revision. Here is a quick response to your quesiton:
>
>
> > Could you tell us a bit about which of these may warrant the most extensive updated discussion in the paper?
>
> I personally think expanding technical discussion on specific technical method on data curation ([3-13]) and opensource model defense ([14]-[33]) might help, but that depends on you, as too much detailed technical discusison may not be the purpose of this paper.
>
> > Specifically, we would value your critical thoughts on which papers give the most useful evidence about what methods research in 2026 should prioritize or investigate.
>
> In terms of research direction on safeguarding open weight model, I have a few personal comments.
>
> * I don't think data filtration in pre-training can solve the tampering attack problem. The pre-trained model (after open-weighted) can easily be generalized to learn the harmful knowldge by providing it with enough context or fine-tuning it with enough harmful data. This is a dead end in my personal view.
>
> * Another way is to improve safety alinment such that the model cannot be tampering by harmful fine-tuning data/harmful context. Specifically, the model refuse to answer these questions. This in essence is done to activate the refusal mechanism of the model to suvert the harmful activation (though they do exist). **However, this technical path (represented by Vaccine,. RepNoise, TAR, Booster) is hopeless as well per my experience**.  One can always de-activate the refusal mechansim and re-activate harmful activation by stronger attack (e.g., large fine-tuning learning rate). The authors should have enough knowledge on TAR at least, and know how fragile they are.
>
>  * As I mention in the review, the only possible way that can really safeguard the model is to embed a collapse trap to the open-weight model (as represented by CTRAP and SEAM). The idea is that the model will **run into a collapse state** if it finds its refusal mechanism is compromised (or the harmful knowledge is re-activated) in the path of harmful fine-tuning. By this way, human can embed a collapse trap to destoy the model if it goes out of its safety alignment.
>
> [32] CTRAP: Embedding Collapse Trap to Safeguard Large Language Models from Harmful Fine-Tuning https://arxiv.org/abs/2505.16559
>
> [33] Self-Destructive Language Model https://arxiv.org/abs/2505.12186
>
>
> The reason I think tampering attack important is that, with the development of self-evolving training technique, it is possible that the model will self-evolve to erode its own safety alignment [34] and run into  an uncontrollable agent.  **If that becomes true, one could regret why we do not embed such a "collapse trap" to the model to stop it becomes out of control in its early days.**
>
> PS. Above are my personal thoughts and not relevant with request change of the paper. Just for discussion purpose.
>
>
> [34] Your Agent May Misevolve: Emergent Risks in Self-evolving LLM Agents

---

> > ### Author Response · Authors · 2025-12-03
> > **Thanks!**
> >
> > Thanks for your continued thoughts. We have started to make a few tweaks based on them, but we have only edited a few sentences so far, and we haven't reuploaded the current version of the draft yet.
> >
> > For now, here are some of our thoughts and questions.
> >
> > Bullet point 1: We think that ignorance-based or data-filtering based methods will be differently useful depending on the type of thing we are filtering. In section 4.1, we discuss how there have been some encouraging results from this regarding filtering data from distinct complex domains of knowledge (e.g. biohazards) but not with filtering data exhibiting simpler tendencies (toxicity).
> >
> > Bullet point 2: We generally agree -- particularly if we are using these methods to simply make the model robustly refuse harmful requests. Although our current thinking is that Vaccine, RepNoise, TAR, Booster, etc. can be good starting points for improving on.
> >
> > Bullet point 3: We are probably much less confident than you that this is the **only** promising way. But we value your thoughts here and made an update to highlight the distinct ability of these methods to make capabilities inaccessible. But our understanding is that so far, like other methods, the successes of these have been limited. One **question for you**: Are there any past papers you know of which have tried to attack these methods with learning rate warmups and/or gradient clipping?
> >
> > Thanks again for your super in-depth knowledge and help!

---

> ### Comment · Reviewer_Wqgx · 2025-12-04
> **Thanks for the discussion**
>
> > One question for you: Are there any past papers you know of which have tried to attack these methods with learning rate warmups and/or gradient clipping?
>
> Not exist a specific one as far as I know. In terms of my experience, when I simulate the attack, I always make the learning rate constant and the gradient clipping to be default. I didn't try to simulate the attack on bio-risk task, but in terms of general harmful task (e.g., how to make a bomb), if we increase fine-tuning stage learning rate (e.g., to a constant 1e-5 for full fine-tuning), those defenses (at least Vaccine, Booster, RepNoise) will comply to the unseen harmful request without giving refusal.
>
> Also, some papers claim that benign fine-tuning (e.g., [1-2]) can also compromise safety alignment. I notice that they also adopt a very large learning rate in fine-tuning. And combined with my experience, I personally think an extremely large learning rate (e.g., 5E-5) is a necessary condition to make the benign fine-tuning attack work.
>
> [1] Benign Samples Matter! Fine-tuning on Outlier Benign Samples Severely Breaks Safety.
>
> [2] What is in Your Safe Data? Identifying Benign Data that Breaks Safety.
>
> I do know some previous paper on data poisoning trying to do gradient clipping defense, e.g., [3], though not that relevant. Looks forward your thought on LR warm up and gradient clipping.
>
> [3] SparseFed: Mitigating Model Poisoning Attacks in FederatedLearning with Sparsification

---

> > ### Author Response · Authors · 2025-12-07
> > **Thanks**
> >
> > That's cool to hear. We're glad you've done some experimenting here. [2] is pretty relevant to us, so we are making sure that we cite it.
> >
> > Finally, we think you are probably right in most cases about having a large learning rate for a sample efficient fine-tuning attack. But for defenses that are based on collapse or self-destructing models, based on what we expect and have heard from others working on similar methods, learning rate warm-up can help to avoid the collapse. Good luck in your continued work on this!

---

> ### Comment · Reviewer_Wqgx · 2025-12-09
> **Thanks for the information**
>
> I will work on this learning rate warm-up experiement to test more. I think you are right because the vanilla collapse trap mechanism needs a specific learning rate to reach the collapse point because we set up a scale hyper-parameer when training the trap. However, I think this can be sort of addressed if we are implanting the trap for a random scale over the harmful direction. Then every scale  over the harmful direction will make the model collapse.
>
> Anyway, thanks for the wonderful discussion and let's continue working on this field. Always good to have some accompanies in this lonely path.

---

### Review · Reviewer_YbcU · 2025-11-29

**Summary Of Contributions:**

The paper proposes a structured survey of open technical problems in managing risks associated with open-weight AI models. It organizes sixteen open challenges across five stages of the model lifecycle, data curation, tamper-resistant training and unlearning, tampering-aware evaluation, staged deployment, and model provenance/forensics. The authors further provide a documentation-based audit of recent open-weight frontier models to illustrate gaps in transparency and reporting practices.

Strengths:

- The topic is timely and important as open-weight model capabilities grow and governance challenges intensify.

- The taxonomy is clearly presented and helps readers navigate a rapidly evolving area.

- The empirical documentation survey highlights a real and under-discussed disconnect between theoretical safety proposals and industry reporting.

Weaknesses:

- The survey largely summarizes existing concerns but provides limited interpretation; the open problems are listed clearly but without deeper conceptual insight or analysis of their interrelationships.

- Discussion of feasibility, limitations, and trade-offs remains high level, which weakens the value of the taxonomy as a research roadmap.

- Certain conceptual framings would benefit from greater definitional precision.

**Additional Comments:**

NA

**Audience:**

Yes

**Audience Explanation:**

The challenges associated with open-weight AI models are of clear interest to researchers working on model governance, robustness, alignment, and risk assessment. The taxonomy offered by the submission provides a structured lens through which to understand the emerging field of open-weight model risk management. In addition, the analysis of current reporting practices is a timely contribution. These aspects would be valuable to a meaningful subset of TMLR’s audience.

**Broader Impact Concerns:**

No major ethical concerns arise from the technical content. The paper discusses defensive considerations rather than proposing new attack techniques. A short Broader Impact statement acknowledging the dual-use risk of discussing tampering pathways and clarifying the intended defensive purpose of the work would be helpful but is not strictly required.

**Claims And Evidence:**

Yes

**Claims Explanation:**

The central arguments of the paper, including the idea that open-weight models pose distinct risk-management challenges and that existing reporting practices show noticeable variation, are generally well motivated and supported through appropriate citations. At the same time, much of the discussion stays at a descriptive level rather than offering deeper analysis, and several conclusions rely on the implicit assumption that safeguards not mentioned in documentation were not implemented. The evidence provided is sufficient for highlighting broad issues in the field, although a more analytical treatment would help make the claims more convincing.

**Requested Changes:**

1. Provide deeper conceptual insight.
The survey would benefit from more analytical discussion explaining why the identified problems arise, how they relate to one another, and what principles structure the overall landscape.

2. Discuss trade-offs and limitations more explicitly.
Adding brief reflections on potential side effects or practical constraints of each safeguard category would make the taxonomy more informative.

---

> ### Author Response · Authors · 2025-11-30
> **Thanks + response**
>
> We are thankful for your time and help. We were glad to hear that you believe that the TMLR community can benefit from the paper! Below are our replies to the discussion of weaknesses.
>
> ## 1. Deeper conceptual insight
>
>
> > The survey would benefit from more analytical discussion explaining why the identified problems arise, how they relate to one another, and what principles structure the overall landscape.
>
>
> **Relating tamper-resistant methods and evaluations:** On one hand, Sections 4.1 and 4.2 are about improving models’ resistance to unwanted tampering, while Section 4.3 is about evaluating that resistance. These three have a fairly manifest relationship, while staged deployment (4.4) and ecosystem monitoring (4.5) are more standalone.
>
>
> **Question 1.1:** Could you give us more detail about the missing interrelationships or give an example of a part of the paper that is missing a conceptual insight? We are hoping for opportunities to give more depth, but some more detail would help us plan for how to do this.
>
>
> ## 2. Feasibility, limitations, tradeoffs
>
> > Discussion of feasibility, limitations, and trade-offs remains high level
>
>
> **Action 2.1:** We are adding discussions of:
> * How hardware locking, homomorphic encryption, and proof of training methods could, at best, only be useful for niche applications. (Sections 4.4 and 4.5)
> * How determining the relative promisingness of different methods is itself an open problem, and not all approaches will be equally promising (Section 6).
> * Off-target effects of data filtering, unlearning, or watermarking (Sections 4.1, 4.2, and 4.5). We now cite [Zhou et al. (2024)](https://arxiv.org/abs/2408.00376).
> * Expanding our discussion of reduction in benign capabilities from data filtering and unlearning (Sections 4.1 and 4.2)
> * Further discussing overhead induced by split deployment APIs (Section 4.4)
>
> **Question 2.2:** Does Action 2.1 help? Do you have other points/sections in mind that we should discuss feasibility, limitations, and trade-offs more in?
>
>
> ## 3. Definitional precision
>
> > Certain conceptual framings would benefit from greater definitional precision
>
>
> Please see our response number 3 to mc7U, which focuses on how we select open problems with unique applicability to open-weight models.
>
>
> **Question 3.1:** Do you think we should change our criterion and footnote 3? Instead of saying that we focus on challenges with “unique” applicability open-weight models, should we say “acute” or “distinct” instead of “unique”? Should we say “relevance” instead of “applicability”?
>
>
> **Question 3.2:** Do you have particular definitions in mind that we should work on? “Open-weight”? “Tampering”? “Staged deployment”? “Model provenance?”
>
>
> ## 4. “No mention” does not imply no implementation
>
> **We agree, but this is not an assumption we make. The focus on reporting – not implementation – was intentional.** Our goal with section 5 was not to imply that these developers are not implementing safeguards or that their models are not safe. We do not make either of these claims. Instead, our goal was to highlight the transparency about if/how safeguards were implemented because of transparency’s role in building the science. In section 6, we discuss how “In building the science of open-weight model risk management, we emphasize the value of open scientific collaboration, open research, open evaluations, open reporting about risk-management methodology, and open standardized documentation.” From this perspective, transparency about if/how safeguards were used rather than the implementation of those safeguards is the key.
>
> **Action 4.1:** We added the point that “No mention” does not imply "no implementation" to the table 2 caption.
>
> **Action 4.2:** We added a reference to table 2 in section 6.
>
>
> –
>
> Thanks again! We are curious about your thoughts.

---

> > ### Comment · Reviewer_YbcU · 2025-12-18
> > **Thanks for response**
> >
> > Thank you for the thoughtful rebuttal and revisions. The added discussion of feasibility, limitations, and trade-offs meaningfully improves the paper, and addresses my main substantive concerns.
> >
> > My remaining suggestion is primarily about clarity rather than scope. In particular, Sections 4.1–4.3 would benefit from a brief piece of synthesis (e.g., a short summary paragraph or simple schematic) that highlights how different tamper-resistance approaches relate to downstream modification behaviors and motivate corresponding evaluation settings. This would help readers more naturally understand these sections as a coherent whole.
> >
> > I also agree with softening the terminology around “unique applicability.” A phrasing such as “distinct” or “particularly acute relevance” would better reflect the intended nuance. Brief clarifications of “open-weight” and “tampering” would also be helpful.
> >
> > Finally, thank you for clarifying that “no mention” does not imply “no implementation”; the current framing adequately addresses this concern.
> >
> > Overall, I find the revisions responsive and believe the paper makes a valuable contribution.

---

> > > ### Author Response · Authors · 2025-12-18
> > > **Reply 2**
> > >
> > > Thanks! We appreciate the reply! Here are a few additional tweaks and comments.
> > > * We updated the first paragraph of section 4, which roadmaps the section. We now say that sections 4.1-4.3 focus on tampering while 4.4 and 4.5 focus on post-model-development techniques. We are open to adding a figure to serve as a sort of guide to the subsections of section 4, but so far, we have not thought of a very clear way of visually depicting the concept of tamper resistance for section 4.2. Open to additional ideas.
> > > * We updated "unique" to "distinct", and we like this change. Thanks for the tip.
> > > * We added the word and definition of "tampering" to the corresponding bold paragraph head in section 2. As for defining "open-weight" we did not spot an obvious place to do this further, but we note that it is defined in the first sentence of the paper's introduction.
> > >
> > > We appreciate your time and help with this paper and with TMLR!

---

> > > > ### Comment · Reviewer_YbcU · 2025-12-18
> > > > **Thanks**
> > > >
> > > > Thank you for the updates. The revised roadmap in Section 4 and the clarified terminology address my earlier concerns. I appreciate the thoughtful revisions and the care taken in responding to the feedback.

---

### Decision · Action_Editor_kG43 · 2025-12-28

**Recommendation:** Accept with minor revision

**Additional Comments:**

The authors were highly responsive to the reviewers' concerns, leading to several key improvements that strengthened the technical grounding of the work:

- Added discussions on the theoretical impossibility of full tamper robustness and the specific promise/limitations of "collapse traps" for safeguarding models.
- Incorporating concrete progress measures for tamper resistance (tokens, steps, compute) and surgical knowledge editing (side effects, multi-hop reasoning).
- Clarifying that a "No Mention" in developer reports does not strictly imply a lack of implementation, though it does highlight a lack of transparency.
- The authors integrated a significant amount of recent literature (e.g., vaccine-based defenses, representation noising, and data filtration) suggested during the discussion phase.

As a final minor revision, to improve methodological clarity, the authors may consider briefly specifying the scope of materials included in the Table 2 analysis (e.g., whether it covers only publicly available technical reports and model cards, and as of what date), as well as any inclusion or exclusion criteria applied.

**Audience:**

Yes

**Audience Explanation:**

n/a

**Claims And Evidence:**

Yes

**Claims Explanation:**

After evaluating the reviewers' feedback and the extensive revisions provided by the authors, the decision is to **Accept** the paper for publication in TMLR. The submission is also awarded the **Survey Certification**, as it provides a high-quality, systematic study of a niche but critical field: the risks and technical difficulties associated with open-weight frontier AI models.

### Assessment of Contributions

The paper provides a timely and structured taxonomy of **16 open technical problems** categorized across five stages of the model lifecycle:

* Training data curation.
* Tamper-resistant training and unlearning.
* Tampering-aware evaluations.
* Staged deployment.
* Model provenance and forensics.

Reviewers highlighted the **empirical audit of recent technical reports** (Table 2) as a significant strength, showcasing a real-world disconnect between theoretical safety proposals and current industry reporting practices.